# Cost-Benefit Analysis versus Cost-Effectiveness Analysis from a Societal Perspective in Healthcare

**DOI:** 10.3390/ijerph20054637

**Published:** 2023-03-06

**Authors:** Robert J. Brent

**Affiliations:** Department of Economics, Fordham University, Bronx, NY 10458, USA; brent@fordham.edu

**Keywords:** cost-effectiveness analysis, cost–benefit analysis, healthcare

## Abstract

Cost-effectiveness analysis (CEA) is the main way that economic evaluations are carried out in the health care field. However, CEA has limited validity in deciding whether any health care evaluation is socially worthwhile and hence justifies funding. Cost-Benefit Analysis (CBA) is the economic evaluation method that should be used to help decide what to invest in when the objective is to record the impact on everyone in society. Cost-utility analysis (CUA), which has its roots in CEA, can be converted into CBA under certain circumstances that are not general. In this article, the strengths and weaknesses of CEA relative to CBA are analyzed in stages, starting in its most classical form and then proceeding via CUA to end up as CBA. The analysis takes place mainly in the context of five actual dementia interventions that have already been found to pass a CBA test. The CBA data is recast into CEA and CUA terms in tabular form in order that the contrast been CEA and CBA is most transparent. We find that how much of the fixed budget that is used up to fund other alternatives determines how much is left over to fund the particular intervention one is evaluating.

## 1. Introduction

Using cost-effectiveness analysis (CEA) in healthcare, by finding the intervention which produces a specific outcome at the lowest cost, may be useful when the perspective is limited to a hospital, healthcare provider, government agency or other healthcare institutional setting, especially when a budget has been provided to fund the least cost interventions. But, when the perspective is social, meaning everyone in society, whether they be family members, third parties or taxpayers is involved, and a budget has not been allocated, a CEA is completely insufficient for deciding priorities as to which interventions, if any, should be funded. In this context, cost-benefit analysis (CBA) is the more relevant economic evaluation method.

The purpose of this paper is to explain precisely why, and how, CBA is the more relevant evaluation method when one is taking a societal perspective in healthcare when evaluating interventions (For a CBA text that focuses solely on the field of heath care, see [1]). This will be carried out mainly in the context of evaluating five new interventions for reducing dementia symptoms. The new dementia interventions involved are: years of education (a dropout prevention program), Medicare eligibility (the extra services it provides), hearing aids (over a lifetime), vision correction (over a lifetime), and avoiding living in a nursing home (as residing in a nursing home increases dementia symptoms). What made these five interventions “new” was not that the medical literature was unaware of them; rather it was because they were recently fully evaluated using CBA, and newly shown to be worthwhile financing, see [2].

For the five dementia interventions we will be highlighting, estimation of benefits and costs were carried out using a large, national panel data set from the National Alzheimer’s Coordinating Center (NACC) (Alzheimer’s is the main category of dementia experienced throughout the US. In our NACC data used for making the economic evaluation calculations presented in this article, we cover all categories of dementia). However, in order that the contrast between using CEA and CBA be fully transparent, some of the data used for the CBA evaluations will be recast also into CEA terms.

The outline of the paper is as follows. We start the analysis by viewing CEA as a method of economic evaluation in its classical form, which involves covering mutually exclusive interventions with, and without, a budget constraint. Then we move on to the more general form of CEA which is called cost-utility analysis (CUA). CUA can be viewed as CEA, but it also can be converted into a CBA with one extension. This leads the analysis to CBA proper which, unlike CEA, can be applied to any type of healthcare intervention to assess whether it is socially worthwhile. In the discussion section, some of the background wider issues concerning CEA and dementia interventions are presented. We close with the summary and conclusions.

## 2. Cost-Effectiveness Analysis

### 2.1. Cost-Effectiveness Analysis with a Budget Constraint

Here we will present the findings of the evaluations of the five new dementia interventions using the CEA methodology. We will assume that the institutional setting involves the US government, say Medicare or Medicaid is making the expenditure decisions (Of the five dementia interventions, only education, Medicare eligibility, and avoiding nursing homes actually involved government involvement).

The starting point for a CEA is to define the effect unit that is to be costed for each of the interventions. For the five new dementia interventions, the effect was the reduction in dementia symptoms as measured by the Clinical Dementia Rating (CDR) scale, known as the *CDR^®^ Dementia Staging Instrument*, created by Washington University (The CDR is a measure of dementia severity used globally that is based primarily on a neurological exam and informant reporting, see [3]. A CDR was administered to each NACC participant at each visit by a clinician. There are six domains in the CDR: memory, orientation, judgment and problem solving, community affairs, home and hobbies, and personal care. Each domain is assessed using a 0 to 3 interval (none, mild, moderate and severe) with a questionable response being scored as 0.5. The CDR-SB (the CDR sum of boxes) is the aggregate score across all six domains and this has a range of 0 to 18. It is important to understand that using dementia symptoms as the effect is basing cognitive impairment on a behavioral definition of dementia (interfering with activities of daily living) rather than the medical definition, which relies on brain pathology (for example fibers and plaques for Alzheimer’s). At this time, there are no interventions that can alter brain pathology, but there does already exist interventions that can reduce dementia symptoms.

To obtain the effect from any intervention, resources have to be allocated and they have to be costed. In CEA the costing is usually related just to the institution undertaking the financing, which is called the private costs. Although the costing for the CBA’s of the new dementia interventions was wider than this, we will assume, just for simplicity, that the CBA costs were the relevant ones for the CEA (We will relax this assumption when we deal with externalities when the analysis switches to social CBA).

Dividing these cost estimates by the outcome effects produces the cost-effectiveness ratios for each intervention. CEA proceeds by ranking each intervention from the lowest to the highest ratio. Table 1 shows how CEA would rank the five new dementia interventions. On the basis of this ranking of the five interventions, education would be the most cost-effective, and avoiding living in a nursing home would be judged the least-cost-effective.

As to the issue of which interventions, if any, would actually be funded, the budget constraint needs to be specified, which involves knowing the amounts available to the government to devote to the interventions. We will consider three possible budget limits: $10,000, $20,000 or $70,000 per person. For a budget of $10,000, the government would only approve education, corrective lenses and Medicare eligibility, as the cumulative sum totals $8705. For a budget of $20,000, hearing aids would be added (making the total $17,2037). For a budget of $70,000 not even avoiding nursing homes, which has by far the largest effect, would be approved, as the total cost with this intervention added would amount to $71,748.

It is clear from this application of the CEA method to the new dementia interventions that the decision-maker who specified how large the budget for the institution was going to be, was effectively determining which interventions were worthwhile for that institution and therefore were going to be carried out. The fundamental weakness of CEA is that this budget decision would be made in advance of knowing what interventions were available, and what the costs and effects are likely to be of the interventions that were available.

### 2.2. Cost-Effectiveness Analysis without a Budget Constraint

In the absence of a pre-specified budget constraint, the main role of CEA as an intervention evaluation method is to indicate which dementia interventions can be eliminated from consideration and therefore not to be financed. There are two categories of interventions that can be eliminated.

The first category contains any interventions that are not effective. For example, take the case of placing an older adult for skilled nursing care in the custody of a nursing home. In the case of the interventions listed in Table 1, if under consideration is living in a nursing home (that is, *not* avoiding residing in a nursing home), Table 1 informs the decision-maker that residing in a nursing home actually increases dementia symptoms by 3.3267 points. This intervention would be counterproductive and so should be eliminated from consideration. In the process, one does not experience the increased dementia symptoms and one obtains a cost saving of $54,545. It is because of this effect gain and the avoidance of incurring costs, that avoiding living in a nursing home becomes a productive intervention and can be included in Table 1.

The second category of intervention that can be eliminated involves an effect gain, or a cost saving from an intervention not listed, but involves some variation of the ones listed. The listed ones are not mutually exclusive as both hearing aids and corrective lenses can both be approved (The listed ones must also be not repeatable, or else one can just fund one intervention over and over again if that is the one that is the most cost-effective). However, if a variation of a listed intervention is the new intervention being evaluated, then the new intervention is mutually exclusive, as both the listed and the new intervention cannot both take place at the same time. In this case, the cost-effectiveness of the particular listed intervention becomes the benchmark for deciding the fate of the new intervention. Thus, for example, consider instead of purchasing five sets of hearing aids over a lifetime, which is what the hearing aid intervention listed in Table 1 involved, one is now evaluating instead purchasing six hearing aids over one’s lifetime. If this sixth set costs more than $8498, yet does not produce greater effects, then it can safely be eliminated as it is dominated by the existing listed intervention (More generally, for mutually exclusive interventions where the new intervention has both a different cost and effect than an existing intervention, the CEA must be decided on the basis of the incremental cost-effectiveness ratios rather than average cost-effectiveness ratios that we have been using. See [4]).

As we have just seen, CEA without a budget constraint can be useful by eliminating some interventions that are not effective, and are dominated by variations of the listed interventions. However, the decision-maker still does not know with CEA whether any of the listed interventions in Table 1 are worthwhile funding.

### 2.3. Cost-Utility Analysis as Cost-Effectiveness Analysis

From the perspective of CEA, Cost-Utility Analysis (CUA) is a CEA that involves the most general health care outcome, which is a Quality Adjusted Life Year (QALY). A QALY is the product of the number of life-years affected (LY) and the quality of any one life year (QoL). In principle, every health care intervention that one can think of, that has an effect, must affect either the quantity or quality (or both) of a person’s life, and that is why it is the most general effect to use in a CEA.

Three of the new interventions listed in Table 1 had CBAs that used data that can be expressed in units of QALYs. In Table 2 we present the three QALY effects, and combine them with the costs from Table 1, to form the cost-utility ratio which is the cost-effectiveness ratio in a CUA.

What is interesting about the alternative cost-effectiveness ratios in Table 2 is that the ratios with QALYs as the measure of effect is much higher than when dementia symptoms were the measure of effect. This confirms the obvious point that CEA ratios very much depend on the specific effect measure it uses. Thus, CEA’s applicability is not general unless it is in the form of a CUA.

What is not so obvious is that an intervention’s chances of being approved is very much dependent on what other interventions are *not* being evaluated. Not appearing in Table 2 are the education and Medicare interventions, because QALY information was not available for these two interventions. Without consideration of these two interventions, Table 2 reveals that avoiding nursing homes would now be approved if the budget were USD 70,000 (as the total cost of the three interventions would be USD 63,808) while before it was rejected. Even with CEA in the form of a CUA, assigning a budget in advance of knowing which interventions will be evaluated, and actually funded, is not a rational economic evaluation method.

When no budget constraint has been assigned, to use CUA for decision-making purposes, CUA league tables are often referred to. These tables rank from lowest to highest, in terms of their cost per QALY, a host of interventions appearing in the literature. These tables are then used for comparison with the particular intervention one is evaluating using CUA, which in our case relates to dementia. Table 3 gives an abbreviated league table that appeared in [5] (The interventions in Mason et al.’s (1993) table are valued in 1990 UK pounds. To facilitate comparison with the US dementia interventions used in this article, which were mainly in 2000 dollars, the UK 1990 GBP value was raised to its 2000 GBP equivalent, using the consumer price index, and then converted to USD using the official foreign exchange rate. Thus, the conversion involved multiplying the GBP amount by 2.1818).

Comparing the cost-utility ratios in Table 2 with those in Table 3, we would conclude that none of the three new dementia interventions was as cost-effective as cholesterol testing and diet therapy, which had the lowest cost-utility ratio of all listed by [5]. However, all three dementia interventions were more cost-effective than Erythropoietin Treatment. Clearly, it matters which intervention in the league table you are using for comparison purposes. Mason et al. are right to point out that for league tables to be valid, they need to be standardized, such that the same measures of utility, cost and discount rate, are used to calculate the cost-utility ratios. But, standardization does not solve the problem of knowing which intervention in the league table is to be used as the benchmark. Just as important is the fact that even when a benchmark intervention has been identified in the CUA league table, one still does not know whether that benchmark is worthwhile or not. A CBA of the benchmark intervention first needs to be undertaken, in order to know whether being more cost-effective than the benchmark justifies funding the intervention.

## 3. Cost-Benefit Analysis

The definition of a benefit is an effect that is valued in monetary terms. Because it is expressed in monetary terms, it is then commensurate with the costs, which are almost always measured in monetary terms (For an exception, where costs and benefits are both expressed in non-monetary terms ((time is the numeraire), see the CBA of the 55-mph speed limit in [6]). It is therefore now possible to compare directly the benefits and costs to see which is greater. If, and only if, the benefits exceed the costs, that is, the difference (called the net-benefits) is positive, then the intervention is worthwhile from the institutional perspective. If the costs and benefits relate to everyone who lives in society, which means that they are *social* benefits and *social* costs, then any positive net-benefits indicate that the intervention is socially worthwhile.

### 3.1. Cost-Utility Analysis as Cost-Benefit Analysis

The effect in a CUA is a QALY (Thus, a QALY is in each individual’s utility function. The social utility is then simply the sum of each individual’s utility function, which is the total number of QALYs for everyone from an intervention). To be designated as a benefit, the QALY must be valued, that is, given a price. In CUA that purports to be a CBA, the price is treated as a constant and is determined independently from the circumstances of the intervention being evaluated. The CUA constant price is a threshold price, usually set at the national level. It is the minimum price assigned to the QALYs in order for the intervention to be judged worthwhile (mainly by others). Often the threshold price is based on some multiple of per capita national income. Ref. [7] surveyed the literature on the threshold value and suggested a QALY price between USD 100,000 and USD 150,000.

As an example of a CUA used as a CBA, refer to [8] evaluation of Cholinesterase Inhibitors and Memantine medicines for those with Alzheimer’s dementia. Their results appear in Table 4. They used the upper limit of USD 150,000 from Neumann et al. as their threshold QALY price. All four monotherapies were found to be socially worthwhile (have positive net-benefits). Since the interventions were mutually exclusive, of the four monotherapies, only Donepezil would have been chosen to be funded.

It is important to understand that the price of a QALY in CBA is primarily meant to be what a person is willing to pay (WTP) for that QALY. This implies that there are two fundamental weaknesses of using a single threshold price to convert a CUA into a CBA. The first weakness, as pointed out by [9], is that it applies a constant price to a QALY. This is a drawback because in economics a demand curve is drawn based on the principle of diminishing marginal utility. This implies that the price that a person is WTP to pay for a QALY decreases as the more QALYS one consumes increases. Johannesson’s point is relevant to the interventions in Table 4 because Donepezil’s total of 1.6 QALYs was greater than for any of the other interventions. If one QALY is valued at $150,000, but the second QALY is valued at only USD 50,000, then the 0.6 additional QALYs is valued at 0.6 of $50,000 and not 0.6 of USD 150,000 as in Table 4. The Donepezil Monotherapy benefits would be downsized to $180,000, making the net benefits become $131,824, which is now lower than for Memantine Monotherapy. Priorities could be altered if the assumption of a constant price is invalid.

The other weakness of CUA using a threshold price as the price of a QALY is that it is not based on a person’s WTP. One of [7]’s justification for the $150,000 threshold came from the World Health Organization’s suggestion that the threshold should be two to three times per capita income, which was around $54,000 in 2014. Using national income as a benchmark is a human capital justification and this is not at all based on an individual’s preferences (The human capital approach used for valuation in CBA in health care assumes that the value of one’s life is the foregone output that society no longer receives because the person dies. The value placed on this foregone output in national income accounting is the price that others place on the products, not what the person whose life is at stake values his or her life).

### 3.2. CBA Not Based on CUA

Whether one selects Donepezil Monotherapy or Memantine Monotherapy from the list of the mutually exclusive rivals in Table 4, its social value does not then need to be compared with any other intervention to be justified. The net-benefits are positive and this is the only prerequisite. This is the criterion for any dementia intervention using CBA to be determined to be socially worthwhile. This means that any type of effect that is valid for an intervention can be used in a CBA and it is not necessary that dementia symptoms be standardized in any way as in Table 1 and Table 2. The effect can be anything that provides value for an individual and society.

In the top part of Table 5 we supply the net-benefits from the CBAs of the five new interventions. Because the effects can be anything that the evaluator considers relevant, the Table is not restricted to the three evaluations that appeared in Table 2 that relied for effects on QALYs (consisting of corrective lenses, hearing aids and avoiding living in nursing homes). Added is education and Medicare eligibility which used the independent living cost savings of reducing dementia symptoms as the effect to be evaluated as in Table 1. It is true that with different measures of effects, the method used for the pricing of effects would be different. But, the point is that if the valuation method used is valid, because it is based on individual preferences, then any intervention with positive net-benefits is worthwhile irrespective of which other alternative interventions are available (providing that they are not mutually exclusive).

The pricing method used for the corrective lenses, hearing aids and avoiding nursing homes interventions was based on the Value of a Statistical Life (VSL) literature. Individual preferences are involved in this method because individuals are willing to trade off a specific probability of dying on the job for the extra salaries that are paid per year to compensate for incurring that extra risk. If a person is willing to accept $5000 as compensation for a one-in-thousand chance of dying, then a thousand times $5000, that is $5 million, is what a thousand times greater risk would be worth, statistically speaking. This simply means that, if a population of 1000 persons are working with a one-in-thousand risk of dying, one should expect, on average, one person would be dying for that $5 million aggregate compensation (The $5 million amount was based on [10]).

The pricing method for the education and Medicare eligibility interventions was in terms of the savings by the effect of reducing dementia symptoms increasing the chances of independent living. When a person can transfer to independent living, caregivers do not have to give up their time and resources looking after the person with dementia. This is true for the government as well as for private citizens as Medicare expenses can go down when people’s dementia symptoms are reduced.

At the bottom part of Table 5 are added two other dementia interventions that did not rely on the NACC data, but illustrate how widespread and multidimensional any dementia intervention can be. Firstly, there is reductions in elder abuse. People take advantage of persons with dementia resulting in psychological, financial and physical elder abuse. This abuse is something that people are WTP to avoid. Using the willingness to prosecute as a measure of this WTP, a benefit amount of between $40,000 to $50,000 was estimated, varying with the type of abuse. By reducing dementia symptoms, one is reducing the extent of elder abuse. Subtracting the cost of $7500 involved with facilitating the prosecution of the abusers, the net-benefits of reducing the dementia symptoms were calculated to be $42,500 (See [2], chapter 8).

The second intervention added to the bottom part of Table 5 was cognitive rehabilitation. Even when dementia symptoms cannot be reduced directly, the consequences of a person’s dementia symptoms can be mitigated, especially for the benefit of a dementia person’s caregiver. Cognitive Rehabilitation, in the form of the Tailored Activity Program (TAP)—see [11]—involves an individual specific intervention whereby an occupational therapist comes to a caregiver house, finds out what dementia behavior needs changing, and trains the dementia person to adapt his/her behavior to reduce the caregiver’s time spent “doing things” or spent “on duty” for the person with dementia. Since time saved by the caregiver can be given a monetary value using labor market valuations, for example, by using the federal minimum wage rate, the benefits of the TAP were straight-forward to estimate. The time saving benefits were put at $8875. The occupational therapist’s time spent traveling to the caregiver’s house, and training the dementia patient and caregiver, was estimated to be $942. Subtracting these costs from the benefits made the net-benefits positive at $7933 (See [2], chapter 9).

The role of Table 5 is to demonstrate the fact that there already exist many dementia interventions that have been evaluated using CBA and found to be socially worthwhile. Many different methods have been employed to put a price on the effect that was found to be the one most relevant by the evaluator of the intervention.

### 3.3. Social CBA

To be a social evaluation, the outcome measure for the CBA cannot be specific to the healthcare institutional setting. The outcome measure must consist of the effects on everyone in society. Similarly, on the costs side, the costs of everyone affected by the intervention must be summed, including those who incur the funding for the intervention. The relevant economic concept here is that on an externality, where one person’s activities affects some other person and this effect is unpriced. Therefore, pricing of the effects on others should be an integral part of a social CBA. If the patient is considered the first party, and the physician or hospital supplying the service to the patient is the second party, then the externality involves the effect on third parties.

In health care, the third party is often the person accompanying the patient to receive the service. The full cost is not just what is charged by the healthcare provider, it is the transport costs and the value of the time given up by the person accompanying the patient. The full benefit is also wider than the gain to the patient, as the friend or family member receives satisfaction when the patient ‘s functioning improves. This externality was explicitly priced in the context of the cognitive rehabilitation intervention referred to in Table 5. The value of the caregiver’s time saved by the TAP constituted the net-benefits of the intervention.

In all the CBAs of the new interventions, the effect was the reduction in dementia symptoms that they produced for the patient, and this led to benefits in terms of either cost savings from increased independent living, or from the value of the QALYs. It is important to understand that when a dementia patient’s symptoms are reduced, this will also mean that the dementia symptoms of the caregiver are also likely to be reduced. This is because the spouses of people with dementia are six times more likely to develop dementia than for persons whose spouses have not experienced dementia [12]. As a result, any reductions in symptoms by the dementia patient will generate external benefits that need to be added to the direct benefits accruing to the dementia patient.

Therefore, all the net-benefit figures for the new interventions in Table 5 must be interpreted to be conservative under-estimates of how socially worthwhile they actually are. The cost-savings from the greater independent living for the caregivers, and the value of the QALYs that they receive externally by the patient’s reduction in symptoms, must be added to the net-benefits of all the new dementia interventions.

Our conclusion that the net-benefits of dementia economic evaluations would likely be higher when externalities are included is supported by the literature. In a survey of 63 CUAs of Alzheimer interventions by [13], for 33 of them they were able to compare CEA ratios with and without externalities. Of these, 28 (85%) had CEA ratios that were either more favorable, or cost-saving, when externalities were included. The main externalities related to informal caregivers in terms of costs (time savings) and QALYs increases they received. For the subset of CUAs that used a threshold price, and thus were converted into CBAs, 21 (64%) of them that did include externalities did not cross the threshold to make the net-benefits negative (The threshold prices they used to value the QALYs were $50,000, $100,00 and $150,000).

## 4. Discussion

### 4.1. Health Evaluation Nomenclature

Although the previous sections have tried to draw a clear line of demarcation line between CEA and CUA, and CUA and CBA, and even CBA from an individual perspective and CBA from a social perspective, the healthcare literature is not careful to label its published work distinctly. One has to actually read a published paper to know whether it is a CEA, or a CUA, or a CBA. The title of the evaluation paper may not be at all informative. For example, even the Yunusa et al. paper, which we analyzed in this article, that applied a QALY price threshold to its CUA results to convert them into CBAs, did not entitle their paper a CUA or a CBA, but instead called it a CEA.

### 4.2. CEA as Cost-Minimization

CEA is most valid, and therefore most useful, if it operates in the context of a fixed quantity of an effect. A CEA would then be a Cost-Minimization (CM) analysis. The outcome of the CEA would identify what combination of resources for an intervention would produce a given effect quantity for the lowest cost. Once this has been determined, CBA can take over and see whether the value of the given effect is greater than the minimum cost combination. As soon as CEA departs from CM by considering also differences in the quantity of effects, and compares this to its differences in costs, it loses its validity. For example, antiretroviral drugs for HIV were the least cost-effective of a number of interventions for HIV in Sub-Saharan Africa [14]; but, when the effect for ARVs was priced to form the benefits, the benefits for this least cost-effective alternative were greater than the costs [15].

### 4.3. CBA and Dementia Interventions

Best practice in CBA is to use a person’s WTP as the price of the effect to estimate the benefits (Costs are usually measured by market prices, with this occasionally adjusted for the extra utility loss to taxpayers for financing the intervention by taxation and other externalities (see [1], part II). For example, for the US at the federal level [16], taxes have on average an extra utility loss of 0.245 per dollar of taxes raised (averaged over all the elasticity possibilities), making a total loss of 1.245. Thus, for any government intervention that is funded by taxes, the costs must be multiplied by 1.245 to form the social costs. Similarly, for any intervention that provides tax savings, the savings must be multiplied by 1.245 to obtain its social value. In the case of Medicare eligibility, all the costs and benefits are in terms of funds involving the government. Thus, the net-benefits of $1,754,545 in Table 5 would be $2,184,409 when the gain in utility from the tax savings is included). This method can be employed for CBAs in healthcare generally, but can be problematic for CBAs of dementia interventions, seeing that WTP to pay depends on ability to pay, and persons with dementia are rarely engaged in paid employment. For the new interventions, the CBAs relied on more indirect WTP measures.

For the education and Medicare eligibility interventions, it was the external benefits that were estimated. The reduction in dementia symptoms led to the older persons being able to shift back to independent living, which produced cost-savings for caregivers and the government.

For the hearing aids, corrective lenses and avoiding nursing homes interventions, the reduction in dementia symptoms generated QALYs which was priced by using the VSL, which is a WTP measure. Although the older persons were not working at the time of the evaluation, the VSL estimates were based on the risk of dying/ extra salary choices by the older persons when they were last working (the VSL amounts used were related to persons aged 62). For the quantity of life years part of a QALY, it was the life expectancy of the older persons that was applied. For the quality of life part of a QALY, it was the dementia person’s stated preferences that were used, as measured by the Geriatric Depression Scale (GDS). In the NACC data set, 95 percent of the patients were judged by trained clinicians to be mentally capable of completing the GDS. Thus, for the GDS, it made sense to use the preferences of dementia patients to help estimate the benefits for these three new interventions. Whenever an economic evaluation uses a person’s preferences to estimate the benefits, it incorporates one of the central value judgments of CBA, which is to honor consumer sovereignty, that is, individuals are regarded to be the best judges of their own welfare.

## 5. Summary and Conclusions

In this article the aim was to highlight the strengths and weaknesses of CEA from a societal perspective as a method of economic evaluation in healthcare. This was carried out in the context of a common, single area of application related to dementia interventions. We started off with a narrow focus, where the effect to be evaluated was restricted just to interventions in terms of dementia symptoms. We then broadened the outcome to consider a comprehensive measure, that of a QALY, that can be adopted for the evaluation of any type of healthcare intervention. Using a QALY converts the CEA into a CUA. Finally, from the perspective of a CBA, which supplies the most valid and general method to use for an economic evaluation, one is not at all limited in the type of effect one employs as long as the effect can be priced.

The applications which we used throughout the analysis concentrated mainly on the interventions that were newly evaluated using CBA and found to be socially worthwhile. These were years of education, Medicare eligibility, hearing aids, corrective lenses, and avoiding living in a nursing home. The data used for these evaluations were recast in order that they could be viewed separately as CEAs, CUAs and CBAs. This enabled the contrast between CEA, CUA and CBA to be fully appreciated. For the CUA part of the analysis, we expanded the range of dementia applications to include FDA-approved medicines. This provided a bridge between CUA and CBA, as not only were QALYs used as the effect of the evaluations, they could also be priced and, in the process, form a special type of CBA. This was because a priced effect is what defines a benefit.

What limited the scope of CUAs from the perspective of CBA generally was that they used a single threshold price that was the same irrespective of the preference of the persons who were actually receiving the benefits of the interventions. When the price of effects was not restricted to a single, threshold price, different methods for estimating the benefits could be employed and this allowed the effects for an intervention to be varied as well. Thus, the list of worthwhile dementia interventions was expanded even further to include the prevention of elder abuse and the provision of cognitive rehabilitation. For CBA evaluations, many different pricing methods can be employed and the two methods we highlighted in this article was in terms of cost savings and the value of a statistical life (Note that although a single VSL amount is adopted, which was Aldy and Viscusi’s $5 million, this does not mean that the valuation of a life is a single price for everyone to whom it is to be applied. This is because, for persons at different ages, the remaining life years varies and this make the valuation of a life year individual specific).

When the effect for the CEA is not a QALY, as when the outcome was considered narrowly to be just the reduction in dementia symptoms, CEA would not be able to be compared with any other healthcare intervention that was not related to dementia. Choices by individuals and by the government in economics are always dependent on opportunity costs. If one does not value alternatives to identify the next best alternative, the opportunity cost of an intervention cannot be determined.

Even when CEA’s contribution in the literature was considered greatest, that is, when non-mutually exclusive interventions were considered, and a budget constraint had been assigned, as an economic evaluation method its role is still very limited, as it is always dependent on the particular alternatives that were identified for comparison purposes. This was because how much of the fixed budget that is used up to finance other alternatives always determines how much is left over to fund the particular intervention one is evaluating. No matter how socially worthwhile an intervention may be, if there are no funds left over to finance it, the intervention will not be approved.

The very existence of a budget constraint can be questioned because it predetermines that something will be funded, even without knowing if any intervention for a specific purpose like dementia was socially worthwhile. More generally, the problem with CEA was that even when funds were available, and an intervention was found to be the most cost-effective one, one still had not accumulated enough evidence to conclude that this low-cost intervention should be approved. An intervention can be cost-effective and not socially worthwhile; or it can be the least cost-effective intervention yet, none-the-less, be socially worthwhile.

On the other hand, whether a budget constraint has been specified (or not) does not limit the use of CBA in any way. When a budget constraint exists, CBA chooses the one with the higher benefit-cost ratio. Without a budget constraint, CBA chooses any alternative with positive net-benefits, as this indicates that this intervention is socially worthwhile.

The other main limitation of CEA relates to its practice. In the economic evaluation literature, it is the costs and effects on the first and second parties to the intervention that are primarily considered. The effects on third parties are usually excluded (This exclusion exists even though there was the recommendation in [17] that states: “All cost-effectiveness analyses should report 2 reference case analyses: one based on a health care sector perspective and another based on a societal perspective”). We emphasized that to be a social evaluation of an intervention, it is the costs and effects on everyone that have be estimated. Externalities need to be included to ensure that it at least becomes a social CEA evaluation. In the context of dementia interventions, it was mainly the costs and effects of the caregivers of the persons with dementia that was the externality that needed to be included.

The main policy prescription that follows from the analysis in this paper is that, when an evaluation is carried out for an intervention in the healthcare field that involves public expenditures, CBA needs to be used as the evaluation method and not CEA. This is because only CBA ensures that outputs will be valued in monetary terms, and therefore made comparable to the costs, to see which is larger, and thereby determine whether the expenditure is socially worthwhile or not. Also, only a CBA provides a social perspective by including the effects on everyone affected by an intervention both directly and indirectly.

## Figures and Tables

**Table 1 ijerph-20-04637-t001:** Cost-Effectiveness of Various Dementia Interventions.

Intervention	Effect: Reduction in Dementia Symptoms	Costs per Person	Cost-Effectiveness Ratio
Education	0.3459	USD 1400	USD 4047
Corrective Lenses	0.1858	USD 765	USD 4117
Medicare Eligibility	0.9182	USD 6540	USD 7123
Hearing Aids	0.7251	USD 8498	USD 11,719
Avoiding Nursing Homes	3.2367	USD 54,545 *	USD 16,892

* The benefit and cost figures for nursing homes in [2] were in population terms. To convert them into per-person terms, one needs to divide by 1.1 million, the approximate number of older adults in Medicaid-financed nursing homes. The total costs of living in nursing homes in the CBA (which was the difference between total benefits of USD 1.93 trillion and the benefits without the nursing home cost savings, which was USD 1.87 trillion) was equal to USD 0.06 trillion. Dividing this sum by 1.1 million produces a cost per person of USD 54,545.

**Table 2 ijerph-20-04637-t002:** Cost-Utility Analysis of Various Dementia Interventions.

Intervention	Effect: Increase in QALYs	Costs per Person	Cost-Utility Ratio
Corrective Lenses	0.1012 *	USD 765	USD 7559
Hearing Aids	0.6785 **	USD 8498	USD 12,525
Avoiding Nursing Homes	3.4282	USD 54,545	USD 15,911

* The outcome measure in [2] for corrective lenses was in terms of mortality and the reduction was 0.0044. Multiplying this mortality reduction by a life expectancy of 23 years produces the equivalent QALY increase of 0.1012. ** The outcome measure in [2] for hearing aids was in terms of the quality of a life year and the reduction was 0.0295. Multiplying this quality of life increase by a life expectancy of 23 years produces the equivalent QALY increase of 0.6785.

**Table 3 ijerph-20-04637-t003:** Cost-Utility Analysis of Various Non-Dementia Interventions.

Intervention	Cost-Utility Ratio
Cholesterol Testing and Diet Therapy	USD 480
Hip Replacement	USD 2545
Kidney Transplant	USD 10,276
Home Haemodialysis	USD 37,658
Erythropoietin Treatment for Anaemia in Dialysis Patients	USD 118,616

**Table 4 ijerph-20-04637-t004:** Net Benefits of Various Dementia FDA-approved medicine interventions.

Intervention	Benefits per Person ^1^	Costs per Person	Net-Benefits per Person
Rivastigmine Oral Monotherapy	USD 131,400	USD 58,277	USD 73,123
Galantamine Monotherapy	USD 144,150	USD 60,793	USD 83,357
Memantine Monotherapy	USD 194,100	USD 48,728	USD 145,372
Donepezil Monotherapy	USD 241,350	USD 48,176	USD 193,174

^1^ The benefit figures in this column were constructed by taking the QALY estimates for each intervention from [8]’s Table 1, and multiplying them by the threshold value of USD 150,000 per QALY.

**Table 5 ijerph-20-04637-t005:** Net Benefits of Various Dementia Interventions.

Intervention	Benefits per Person	Costs per Person	Net-Benefits per Person
Education	$5500	$1400	$4100
Corrective Lenses	$14,249	$765	$13,484
Medicare Eligibility	$9338	$6540	$2798
Hearing Aids	$248,425	$8498	$239,927
Avoiding Nursing Homes	$1700,000	($54,545)	$1754,545
Preventing Elder Abuse	$50,000	$7500	$42,500
Cognitive Rehabilitation	$8875	$942	$7933

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
