# Peer review of "Cost-Benefit Analysis versus Cost-Effectiveness Analysis from a Societal Perspective in Healthcare"

_ijerph, 2023, doi:10.3390/ijerph20054637_

Round 1

Reviewer 1 Report

It is good work but needs some areas of attention, the detail is in the attached report.

Reviewer 2 Report

#1. Conclusion “The other main limitation of CEA relates to its practice. In the economic evaluation literature, it is the costs and effects on the first and second parties to the intervention that are primarily considered. The effects on third parties are usually excluded” >> Please note that “All cost-effectiveness analyses should report 2 reference case analyses: one based on a health care sector perspective and another based on a societal perspective” was the recommendation by the Second Panel on Cost-Effectiveness in Health and Medicine [JAMA. 2016;316(10):1093-1103]. Furthermore, CEA and CUA were still currently valid forms of health economic evaluation in CHEERS 2022. [BMJ . 2022 Jan 11;376:e067975] 

#2. Table 1 "$54,545" >> should it be "(-)$54,545" or “($54, 545)” [ie, (avoiding NH) was less costly than NH] because section 2.2 stated "a cost saving of $54,545. (also see Brent 2022b p 3748) ? If so, the “$16,892” should be replaced as “dominant”? The same issue applied for table 2. It looks like the correct format “($54, 545)” had been used in table 5.

Author Response

Please see the attchment

Round 2

Reviewer 1 Report

It is good revision as required